# Vaccination Open Day: A Cross-Sectional Study on the 2023 Experience in Lombardy Region, Italy

**DOI:** 10.3390/ijerph21060685

**Published:** 2024-05-27

**Authors:** Pier Mario Perrone, Simona Scarioni, Elisa Astorri, Chiara Marrocu, Navpreet Tiwana, Matteo Letzgus, Catia Borriello, Silvana Castaldi

**Affiliations:** 1Department Biomedical Sciences for Health, University of Milan, 20133 Milan, Italy; simona.scarioni@unimi.it (S.S.); elisa.astorri@unimi.it (E.A.); chiara.marrocu@unimi.it (C.M.); silvana.castaldi@policlinico.mi.it (S.C.); 2Fondazione IRCCS Ca’ Granda Ospedale Maggiore Policlinico, 20122 Milan, Italy; navpreet.tiwana@policlinico.mi.it (N.T.); matteo.letzgus@policlinico.mi.it (M.L.); 3Vaccination Unit, ASST Fatebenefratelli Sacco, 20131 Milan, Italy; catia.borriello@asst-fbf-sacco.it; 4Directorate General for Health, Lombardy Region, 20124 Milan, Italy

**Keywords:** influenza vaccination, vaccine, vaccination coverage

## Abstract

Background: Vaccination is a highly effective tool for controlling infectious diseases, particularly in populations at high risk of contagion due to clinical conditions or occupational exposure, such as healthcare workers. The purpose of this study is to present the open day event that marked the beginning of the influenza and anti-COVID-19 vaccination campaign in the Lombardy region and to describe the experience of an Istituto di Ricovero e Cura a Carattere Scientifico in Milan. Methods: During the vaccination open day, eligible individuals received free vaccinations for influenza, COVID-19, pneumococcal disease, and shingles, as provided by the Lombardy Agenzia per la Tutela della Salute. In celebration of the centenary of the Università degli Studi di Milano, the Fondazione Ca’Granda Ospedale Policlinico, a contracted hospital of the university, created a special electronic diary for a total of 150 individuals, equally divided between children aged 2–6, pregnant women, and university staff. Results: At the regional level, a total of 6634 influenza vaccines, 2055 anti-COVID-19 vaccines, 108 anti-pneumococcal vaccines, and 37 anti-zoster vaccines were administered. A total of 3134 (47.3%) influenza vaccines, 1151 (56%) anti-COVID-19 vaccines, and 77 (62%) anti-pneumococcal vaccines, were given to individuals aged 60–79. No differences were observed between the total number of male and female vaccinees (1017 and 1038, respectively), who received the anti-COVID-19 vaccine. At the Policlinico Foundation, out of 150 available booking slots, 154 vaccines were administered, including 117 influenza vaccines. Conclusions: The establishment of vaccine open days is a beneficial way to increase vaccine compliance. Co-administration of little-known vaccinations outside of healthcare settings could also be a useful tool.

## 1. Introduction

Seasonal influenza is a vaccine-preventable illness caused by an orthomyxovirus with significant implications for healthcare systems [1,2,3]. Caused by the influenza virus, an RNA virus. The most prevalent types responsible for seasonal flu epidemics are Influenza A and B. Individuals infected with the influenza virus typically experience a range of symptoms, including fever, chills, muscle aches, cough, congestion, runny nose, headaches, and fatigue. In severe cases, the flu can lead to pneumonia, bronchitis, sinus infections, and the exacerbation of chronic medical conditions [4]. According to estimates from the World Health Organization, seasonal flu is responsible for around one billion cases of illness worldwide, including 3 to 5 million severe diseases and 290,000 to 650,000 deaths annually [5]. At the same time, in Europe, seasonal influenza leads to up to 50 million symptomatic cases and 15,000 to 70,000 deaths [6,7].

In addition to seasonal influenza, however, there are several respiratory diseases with an important health impact that can be prevented by vaccination, in particular the coronavirus disease 2019 (COVID-19) and pneumococcal infections caused by Streptococcus pneumoniae (SP).

COVID-19 is caused by Severe Acute Respiratory Syndrome coronavirus 2 (SARS-CoV-2, previously 2019-nCoV), an enveloped, positive-sense single-stranded genomic RNA virus (+ssRNA). This RNA virus belongs to the coronavirus family and can result in a wide spectrum of symptoms. Common symptoms include fever, cough, shortness of breath, fatigue, muscle or body aches, headache, new loss of taste or smell, sore throat, congestion, nausea, and diarrhea. The severity of COVID-19 varies, with some individuals experiencing mild symptoms, while others develop severe conditions such as pneumonia, acute respiratory distress syndrome (ARDS), multi-organ failure, and death.

SP infections are one of the major causes of morbidity and mortality worldwide. Pneumococcus is one of the principal etiological agents of CAP, bacterial meningitis, otitis media, and chronic obstructive pulmonary disease (COPD) exacerbations. All age ranges are involved, but it is the elderly and children who are particularly high risk [8]. The treatment of pneumococcal infections is complicated by the worldwide emergence in pneumococci of resistance to penicillin and other antibiotics. Pneumococcal disease is preceded by asymptomatic colonization, which is especially high in children [9].

The impact on healthcare systems extends beyond clinical aspects and includes the economic burden of the disease. This element has resulted in expenses associated with hospitalization and lost workdays [10,11,12,13]. Furthermore, complications like pneumonia or congestive heart failure, as well as long-term hospital admissions that could lead to hospital-acquired infection, can exacerbate morbidity and raise costs [14,15].

These significant, clinical and economic repercussions, combined with widespread access to vaccines, establish vaccination as the most effective preventive measure [13,16,17,18,19,20,21,22,23,24,25]. In Italy, influenza vaccination is recommended for healthcare professionals, the hospitalized elderly, those over 60, and young children aged between 6 months and 6 years old [26]. A high vaccination coverage rate (VCR) is crucial in these populations, as highlighted by international health organizations [27]. However, several studies conducted worldwide have shown significantly lower levels [4,28,29,30,31,32,33,34,35,36,37,38]. To achieve adequate coverage in Italy [39], the influenza vaccine could be administered at hospitals or tertiary vaccination centers, as well as in general practitioners’ (GP) studies and pharmacies [40,41]. Trusting relationships between GPs and patients, particularly in cases of chronic illness [24], are a factor influencing vaccination acceptance as identified in the literature [42,43,44,45]. Furthermore, vaccination among healthcare workers and those undergoing long-term hospitalization within a hospital setting could improve adherence [46]. The ease of acquiring vaccinations at work or during work breaks, along with the assurance of a safe environment, accounts for the high uptake [47,48].

The promotion of vaccinations has emerged as a key method to increase VCR [49,50]. Various studies have examined the efficacy of tailored messaging in improving adherence rates [31,51,52,53,54,55]. The importance of advertising, combined with the need for widespread dissemination, led to the development of the vaccination open days. These events are based on the idea of vaccinating a small sample from each category, thereby encouraging them to become vaccine advocates among their colleagues or acquaintances. To launch the influenza vaccination campaign in Lombardy, the Lombardy Regional Welfare Directorate organized a vaccination open day on 1 October 2023, carried out in partnership with the hospital’s networks called Aziende Socio-Sanitarie Territoriali (ASST) and the tertiary healthcare network Agenzie di Tutela della Salute (ATS).

The University of Milan led the initiative for the City of Milan, seizing the opportunity of the University’s centenary to integrate the initiative into the centenary celebrations. The possibility of holding a region-wide open day in addition to the involvement of the University of Milan was made possible by the extensive vaccine distribution network in the Lombardy region.

The aim of this publication is to describe the organization, implementation, and results of the campaign, which was the first of its kind in Italy, and to evaluate the participation of the target population and the impact of the open day on the overall vaccination campaign.

## 2. Materials and Methods

The organization of Vaccination Day was extremely variable due to the difference in the providers among the different provinces and regional ATS. The intrinsic characteristics of the territory, including the presence of a decentralized supply network with respect to a single center, the availability of personnel to be assigned to extraordinary work given the provision of the service on public holidays or the possibility of autonomous transport by patients belonging to a center has in fact made a common organization on the territory very complicated.

Due to the need for greater control and safety in the dispensing procedures, the presence of two operators per vaccination line has been guaranteed reaching a total of about 15 people vaccinated every hour.

On 1 October 2023, each ASST was requested to arrange vaccinations, ideally in the vaccination centers located in socio-health structures spread across the territory (“Casa di Comunità”), with a minimum of 150 reservations and 4 h slots. The open day of the vaccination campaign will target the following groups: people over 60 years of age, children between the ages of 2 and 6, health workers, pregnant women, and other categories of public servants. Access to flu vaccination was available by registering at the designated regional website. The ATS checks the published agendas visible to citizens and determines the appropriateness of the proposed offer. The event was promoted through a specific communication campaign via the web and social media, as well as a press conference organized by the Welfare Department of the Regione Lombardia.

As part of the influenza vaccination campaign, citizens who are eligible due to their age or medical conditions will be given the opportunity to receive either the anti-pneumococcal vaccine or the anti-herpes zoster vaccine. On 1st October, following the arrival of the new anti-COVID-19 vaccines updated to the XBB variants and the note from Regione Lombardia, citizens in Lombardy region who wanted could receive both the anti-COVID-19 and influenza vaccines.

In addition, the distribution of vaccines at the territorial level has followed two logics also based on organizational and logistical complexities. Due to the storage temperature and the need to guarantee the particular cold chain for transport and storage, anti-COVID vaccines required widespread distribution to individual vaccination centers starting from a central national hub. Flu vaccines, on the other side, due to their relative ease of transport and the possibility of being stored at much higher temperatures than anti-COVID vaccines, have been purchased autonomously by the individual centers according to agreements stipulated at regional level with each pharmaceutical company.

With regard to the vaccine types and brands used, a mixed strategy was employed due to the differing compositions of the populations to which the open day was dedicated.

With regard to the anti-pneumococcal vaccination, a single formulation based on a conjugated, adsorbed polysaccharide vaccine containing polysaccharides of 20 different (Apexvax©) serotypes was used. With regard to anti-COVID-19 vaccination, an mRNA vaccine directed against the Omicron XBB variant was used, administered from the age of 12 years (Comirnaty XBB©). Herpes Zoster vaccination was carried out using an adjuvanted recombinant vaccine available in a single formulation (Shingrix©). With regard to influenza vaccination, on the other hand, five vaccine types were used, depending on the clinical and demographic characteristics of the patient. For patients between the ages of 2 and 18, where the use of a spray formulation was deemed appropriate due to the difficulty of administration by intramuscular injection, a live attenuated vaccine with transnasal administration (Fluenz Tetra©) was employed. Subjects over the age of 65 were administered an inactivated, adjuvanted vaccine (Fluad©). Patients over six months of age who could be vaccinated by intramuscular injection were given a split, inactivated, quadrivalent vaccine (Vaxigrip Tetra©). An inactivated vaccine produced in cell cultures (Flucelvax Tetra©) was administered to workers in health and social care facilities. Finally, a vaccine based on a split, inactivated virion (Efluelda©) was administered to individuals over 60 years of age with significant chronicities. As far as technical management is concerned, the use of software that has been in use for some years now and the selection of personnel already adequately trained in the procedure have made it possible to reduce technical difficulties to mere problems due to internet connection problems or computer malfunctions. The software used to manage the registration of the vaccinations was the one created in 2021 to perform the first COVID-19 vaccination Campaign. All the vaccinations were recorded on a dedicated portal, and each citizen could receive a maximum of two administrations on the same day. Through the dedicated monitoring portal, vaccination data were obtained from the first day of vaccination on 1 October 2023.

Information on available and occupied places was collected for influenza vaccinations. However, for anti-COVID-19, anti-pneumococcal, and anti-herpes zoster vaccines, co-administration with the influenza vaccine is the only planned, without any specific agenda. With the exception of the Università degli Studi di Milano, which managed a 150-slot agenda outside the portal and reserved it exclusively for university staff, all vaccination centers used the dedicated portal to schedule vaccinations.

At the same time information on the administration of influenza, anti-COVID-19, an-ti-pneumococcal, and anti-herpes zoster vaccines was also collected. Information on the type of vaccine administered, the category of person vaccinated, the site of vaccination, and the pregnancy status were collected, and data on the age group of those who received the anti-COVID-19 vaccine were also documented. All vaccinations were registered through a dedicated portal, and the ASSTs were tasked with verifying the accuracy of the numbers reported through the monitoring portal and the vaccinations that were administered.

## 3. Results

Table 1 shows the demographic data of the Lombardy region that can be associated with greater or lesser vaccination propensity. Approximately one-third of the population (30.1%) is over 60 years of age with an equal percentage of university graduates (30.6%). With regard to the educational attainment of the population aged over 18 years old, the proportion of individuals in Lombardy holding a higher diploma (65.5%) is almost twice that of university graduates. As far as gender is concerned, no large differences are observed, with the male and female populations corresponding to 50.9% and 49.1%, respectively.

Table 2 shows the reservations made by all the ATSs in Lombardy, categorized according to the eligible groups who had access to the vaccination during the open day. These groups included children aged between 2 and 6 years, pregnant women, healthcare workers, and persons aged over 60 years.

The highest number of reservations was recorded for people aged over 60 years, particularly in ATS Milano. Approximately 33% of appointments were made at ATS Milan, and approximately 30% of them were for people over 60. This correlation is linked to the higher population density of the area covered by this ATS.

In total, 5830 appointments were made in the whole of Lombardy out of the 9557 available places, which results in a saturation rate of 61%.

A total of 14,584 doses were given during the first open day in Lombardy in October.

As described in Table 3, a total of 2055 COVID-19 vaccines were administered during the open day. A total of 1151 (56%) were administered to persons aged between 60 and 79 years. A total of 1195 (58%) vaccines were administered in the territorial vaccination centers, 702 (34%) in the “Case di Comunità” and the remaining 158 (8%) in the vaccination clinics of the hospital. Our results show that there is no significant association between the sex of the subjects, indicating that males and females have a similar level of compliance, as shown in Table 2.

Table 4 shows data about booking systems and data about the ATS population. The average rate of reservation ratio was 61% with slight variation excluding the ATS Brianza which accounts for a reservation ratio close to 97%. The opposite occurs when the availability ratio is considered. Due to the high population of ATS Brianza (1.218.378) and the low number of rounds available (342) that explain the 0.28 per thousand people.

Table 5 shows the results of a single ATS organization. To ensure regular activity in each line, a doctor and a nurse were enlisted at least taking an excess of HCWs in order to guarantee the administration activity prosecution even if in potential need to close a line. Considering nurses, the excess of HCW compared to the line opened could be explained also due to the necessity of operators intended for vaccine preparation.

During the open day, 37 doses of vaccine against the herpes zoster virus were administered as shown in Table 6, which also describes the distribution among the ATSs of Lombardy. It can be seen that about 60% of the administrations took place in the ATS of Milan and Insubria alone. This is confirmed by the general numbers of vaccinations of the ATS of Milan, which are always higher than the regional average, as opposed to the ATS Insubria, which has lower numbers than the Brescia area.

As highlighted in Table 7, a total of 6634 influenza vaccines were administered, of which 1.214 were administered to children aged 0 to 19, 1535 to adults aged 20 to 59, and 3885 to people aged 60 and older. A total of 107 anti-pneumococcal vaccines were administered, of which 101 were to people aged 60 and older.

Table 8 shows a comparison between the flu and anti-pneumococcal vaccine for each ATS. It can be noticed considering the anti-flu vaccine that Milano, Brescia, and Insubria are the ATS that have carried out more vaccinations (67.1%), and while evaluating the anti-pneumococcal vaccine it can be found ATS Bergamo replaced Insubria accounting with Milano and Brescia for the 80.1% vaccination performed.

Table 9 shows a comparison between vaccine administration performed in the first week of the campaign. Despite the impossibility of a direct comparison between 2021/2022/2023 due to the introduction of vaccination day during the 2023/2024 campaign, a high number of administrations could be observed not only for the first day, specifically the vaccination day, but also for each day during the first week. Comparison was performed for both the flu and anti-pneumococcal vaccine, obtaining in both cases the same result.

The IRCCS Ca’ Granda Ospedale Maggiore Policlinico Foundation provided the opportunity to co-administer influenza and COVID-19 vaccines during the open day and this was the first event for the celebration of the first centenary of the University of Milan. As shown in Table 8, a total of 154 vaccines were administered, of which 37 (24%) were COVID-19 vaccines and 117 (76%) were influenza vaccines.

Regarding flu vaccine administrations, the highest rate of compliance was observed among academic staff, as 43 doses (36.8%) were administered to them. Table 10 provides a summary of the categories of persons who received the doses of the different types of vaccines and also lists the specific vaccines that were administered to each category.

## 4. Discussion

Vaccination against influenza, COVID-19, and pneumococcal diseases is essential for individual and public health. These vaccines help prevent infections, reduce the severity of illnesses, and protect high-risk groups from serious complications. Maintaining high vaccination rates is crucial for controlling these diseases and minimizing their impact on healthcare systems and society.

Given the importance of maintaining a high VCR in the fight against influenza, the experience of the regional vaccination open day can be considered a cornerstone. Although it was only a one-day event, the 61% compliance rate highlights the attention given to this preventive measure. About this, Ministry data show a steady increase in CVR over the years, from 16.8% in 2019/2020 to 23.7% in the 2020/2021 campaign [56]. The open day for the 2023/2024 campaign confirms an upward trend. However, our study group will have to wait for the final results to substantiate any significant findings.

However, while a 61% adherence rate is admirable, it still falls short of both the minimum coverage target of 75% and the recommended target of 90% set by the WHO and Italy [57].

This discrepancy could be partially explained by the chosen day and the fear of side effects of vaccination. Indeed, a study conducted on health workers in an Italian hospital showed that 13.1% cited fear of side effects as the reason for refusing to vaccinate [32]. Moreover, the day chosen for the open day was a Sunday and this, combined with the fear of side effects on the following working days, might therefore be one of the reasons for a lower-than-desired adherence. At the same time, however, the possibility of carrying out a vaccination in an environment perceived as safe due to the presence of health workers or because it is seen as a place for emergency response in the event of serious reactions is one of the motivations behind the idea of a vaccination open day. Reducing the fear of side effects could in fact be the basis for greater acceptance of vaccination.

Considering the age variable, despite the greater adherence to the campaign by the over-60-year-old population (60.3%) [58], approximately 18.3% of the doses were administered in the 2–6-year-old population. The focus on vaccination in this population, attention on the part of parents or caregivers, has in fact been widely analyzed in the literature [29,59,60,61]. For this reason and in order to achieve greater compliance on the part of the pediatric patient as well as for their greater efficacy [62,63], trans nasal spray vaccines have been used in this population [61].

At the same time, considering the 60-year population was the wider population during this open day, as mentioned the possibility of requesting the vaccine against the herpes zoster virus was actively offered to eligible patients. The low number of administered doses could be explained by the lack of knowledge about this vaccine and even more about the lack of knowledge about the possibility of being vaccinated among the eligible age group population.

With regard to the demographic data, it should be noted that although the Lombardy region cannot be considered an ideal model for the whole of Italy because of its demographic characteristics, it is an ideal model for a vaccination campaign targeting the most vulnerable groups. This reinforces the importance of this intervention as a tool that can be extended both nationally and, even more so, internationally. At the same time, the possibility of administration of not more than two doses combined to influenza and pneumococcal pneumonia and greater publicity could be considered as the other reasons for such a low adherence rate. Patients could be more interested in influenza and pneumonia vaccination and choose to delay the zoster vaccination.

As mentioned before during the open day experience, the COVID-19 vaccine was administrated. In this regard, by this experience, two elements could be highlighted. The first one is linked to the vaccination center with the high administration rate. This could be justified by the preventive mission of territorial vaccination centers compared to a more clinical vision centered on acute illnesses such as acute pneumonia at hospital vaccination centers. The second one is linked to the change in gender prevalence among the 20–59-year-old group and the 60–89-year-old one. Despite the absence of a clear motivation, it could be noticed that there is a prevalence of female patients in the first group that switch to a male prevalence in the second one. It is interesting to note that despite several studies in the literature trying to evaluate gender differences in COVID-19 vaccination acceptance or hesitancy, no one shows an age-specific break line [64,65,66,67,68].

Lastly, on the occasion of the open day and the centenary of its foundation, the University of Milan also opened an agenda aimed at university staff to promote vaccination in this population. In particular, it was seen that this population was the largest in terms of flu vaccine administration (27.9%), being less interested in COVID-19 vaccinations (16.2%). In this regard, the demand for the COVID-19 vaccine is an obvious issue. Despite the availability of co-administration for every person during the open day indeed only 24% of the eligible population has applied. This represents a major issue in terms of vaccine hesitancy, highlighting two possible problems closely linked to vaccination, namely the misperception of efficacy on the one hand and the decline in interest in those diseases perceived as non-emergency on the other [69].

Think about the importance of open day vaccination, the absence of literature on similar experiences and their influence on vaccination campaign trends explains the interest in this specific topic and the possibility of increasing VCR through these operations. The engagement of territorial centers as well as university hospitals could lead to the comparison between these two entities and the influence of open day on the vaccine campaign continuation.

## 5. Conclusions

Despite not having a specific vaccination campaign, the experience of open day could be considered a first step in order to increase vaccination adherence through the resonance that this event might have. Coadministrations were offered to eligible populations obtaining a pretty good result, especially on pneumococcal vaccines highlighting the importance of this kind of advertising initiative.

At the same time, this experience shows the necessity of rethinking the overall organization through the possibility of a specific vaccine typology open day in order to guarantee an extended averting about the necessity of vaccination and to ensure the possibility of performing vaccination without a complex booking and administration path. In this regard, one of the limitations of this study is the absence of a survey or a centralized feedback collection system for healthcare workers or patients. The possibility of a system of active collection of critical issues or advantages of this mode of disbursement would in fact represent an excellent tool to evaluate the usefulness or reproducibility on a large scale, geographical or temporal, of this experience.

Finally, the university setting experience could be a first step in a process that will lead to an increase in people never vaccinated previously due to age or other reasons that could be involved by an administration performed in their work setting.

In conclusion, as described in the literature, several factors bear different weight based on the context and times of immunization campaigns, but the open day experience could be a new driver to increase interest and acceptance of this.

## Figures and Tables

**Table 1 ijerph-21-00685-t001:** Demographic data about Lombardy region population. *n* = 9,976,509.

	*n* (%)
Sex	
M F	4,900,520 (49.1%) 5,075,989 (50.9%)
Age	
0–19 20–39 40–59 60–79 80+	1,767,970 (17.7%) 2,133,869 (21.4%) 3,072,551 (30.8%) 2,243,019 (22.5%) 759,100 (7.6%)
Education (>18 yr)	
Undergraduate Graduated	6,534,613 (65.5%) 3,052,812 (30.6%)

**Table 2 ijerph-21-00685-t002:** Open day reservations.

ATS	Other Categories	Children 2–6 Years	Pregnant Women	Healthcare Workers	People Aged > 60 Years	Total
ATS BERGAMO	105	76	23	29	458	691
ATS BRESCIA	143	99	13	43	654	952
ATS BRIANZA	73	48	16	20	173	330
ATS INSUBRIA	84	210	20	48	469	831
ATS MILANO	302	460	113	39	1211	2125
ATS MONTAGNA	18	37	0	14	109	178
ATS PAVIA	34	75	6	20	147	282
ATS VALPADANA	56	60	8	23	294	441
Total	815	1065	199	236	3515	5830

**Table 3 ijerph-21-00685-t003:** Anti-COVID vaccine administration per age and sex.

Age	Female (%)	Male (%)	Total (%)
0–11	0	1 (0.1%)	1 (0.0%)
12–15	1 (0.1%)	0	1 (0.0%)
20–29	25 (2.5%)	9 (0.9%)	34 (1.6%)
30–39	98 (9.6%)	48 (4.6%)	146 (7.1%)
40–49	72 (7.1%)	60 (5.8%)	132 (6.4%)
50–59	136 (13.3%)	92 (8.9%)	228 (11.1%)
60–69	295 (29%)	360 (34.7%)	655 (31.9%)
70–79	205 (20.2%)	291 (28.0%)	496 (24.1%)
80–89	146 (14.4%)	152 (14.6%)	298 (14.5%)
90–99	39 (3.8%)	25 (2.4%)	64 (3.1%)
Total	1017	1038	2055

**Table 4 ijerph-21-00685-t004:** Data about ATS booked reservations compared to population and available rounds.

ATS	Reservation	Availability	Population	Availability/1000
ATS BERGAMO	691	1.524 (45.3%)	1,122,936	1.36
ATS BRESCIA	952	1.400 (68.0%)	1,177,932	1.19
ATS BRIANZA	330	342 (96.5%)	1,218,378	0.28
ATS INSUBRIA	831	1.506 (55.2%)	1,476,843	1.02
ATS MILANO	2.125	3.222 (65.9%)	3,549,500	0.91
ATS MONTAGNA	178	349 (51.0%)	293,711	1.19
ATS PAVIA	282	550 (51.3%)	547,994	1.00
ATS VAL PADANA	441	664 (66.4%)	766,873	0.87
Total	5830	9557 (61.0%)	10,154,167	0.94

Data in the reservation column indicates the number of reserved slots in the vaccination sessions. The availability column shows the number of available slots, with the occupancy rate calculated as the ratio of availability to reservation. Finally, the column availability/1000 shows the ratio of available slots to population per thousand subjects.

**Table 5 ijerph-21-00685-t005:** Data about Lombardy ATSs logistic organization.

ATS	Line Opened	Physician	Nurse
ATS BERGAMO	14	16	16
ATS BRESCIA	19	20	20
ATS BRIANZA	7	8	8
ATS INSUBRIA	17	20	20
ATS MILANO	43	50	50
ATS MONTAGNA	4	6	6
ATS PAVIA	6	7	7
ATS VALPADANA	9	10	10
Total	815	1065	1065

**Table 6 ijerph-21-00685-t006:** Anti-herpes zoster virus vaccine administrations.

Site	Administrations
ATS BERGAMO	3 (8.1%)
ATS BRESCIA	9 (24.3%)
ATS INSUBRIA	10 (27.0%)
ATS MILANO	12 (32.4%)
ATS PAVIA	1 (2.7%)
ATS VAL PADANA	2 (5.4%)
Total	37

**Table 7 ijerph-21-00685-t007:** Anti-flu and anti-pneumococcal vaccines administrations by age.

Age	Flu Vaccine Administrations (%)	Anti-Pneumococcal Vaccine Administrations (%)
0–11	1197 (18.0%)	0
12–15	14 (0.2%)	0
16–19	3 (0.1%)	0
20–29	98 (1.5%)	0
30–39	501 (7.6%)	0
40–49	414 (6.2%)	1 (0.9%)
50–59	522 (7.9%)	5 (4.7%)
60–69	1762 (26.6%)	46 (43.0%)
70–79	1372 (20.7%)	31 (29.0%)
80–89	651 (9.8%)	20 (18.7%)
90–99	100 (1.4%)	4 (3.7%)
Total	6634	107

**Table 8 ijerph-21-00685-t008:** Anti-flu and anti-pneumococcal vaccine administrations by ATS.

ATS	Anti-Flu Vaccine	Anti-Pneumococcal Vaccine	Total
ATS BERGAMO	759 (11.4%)	31 (29.0%)	790 (11.7%)
ATS BRESCIA	1105 (16.7%)	18 (16.8%)	1123 (16.7%)
ATS BRIANZA	355 (5.4%)	3 (2.8%)	358 (5.3%)
ATS INSUBRIA	1002 (15.1%)	2 (1.9%)	1004 (14.9%)
ATS MILANO	2343 (35.3%)	45 (42.1%)	2388 (35.3%)
ATS MONTAGNA	230 (3.5%)	0	230 (3.4%)
ATS PAVIA	323 (4.9%)	7 (6.5%)	330 (4.9%)
ATS VALPADANA	517 (7.8%)	1 (0.9%)	518 (7.7%)
Total	6634	107	6741

**Table 9 ijerph-21-00685-t009:** Administration performed during first week of vaccination in 2021, 2022, and 2023.

Day of Administration	2021	2022	2023
Flu vaccine			
01/10	1	17	6635
02/10	2	69	633
03/10	3	35	727
04/10	14	55	977
05/10	5	1373	1880
06/10	115	1745	3158
07/10	1981	3342	1198
Anti-pneumo vaccine			
01/10	23	1	108
02/10	2	0	50
03/10	0	11	148
04/10	35	21	27
05/10	27	24	151
06/10	39	24	208
07/10	10	59	59

Day of administration report the date expressed as the day and month of the first week of the flu vaccination campaign in the years 2021, 2022, and 2023. The columns 2021, 2022, and 2023 report for each day the number of influenza vaccines carried out on the day.

**Table 10 ijerph-21-00685-t010:** Anti-flu and anti-SARS-CoV-2 vaccines administrations in IRCCS Ca’ Granda Ospedale Maggiore Policlinico Foundation.

Vaccine	Number of Doses (%)
ANTI-FLU VACCINE	117
Healthcare workers	7 (6.0%)
Academic staff	43 (36.8%)
People > 60 years	21 (17.9%)
Individuals at risk due to medical conditions	6 (5.1%)
Age between 6 months and 6 years	34 (29.1%)
Pregnant women/postpartum	5 (4.3%)
Cohabitant of an individual at risk	1 (0.9%)
ANTI-SARSCOV2 VACCINE	37
Cohabitant of an individual at risk	1 (2.7%)
Age < 60 years	8 (21.7%)
Age > 60 years	14 (37.8%)
Healthcare workers	6 (16.2%)
Academic staff	6 (16.2%)
Individuals at risk due to medical conditions	2 (5.4%)
Total	154

## Data Availability

The data presented in this study are available on request from the corresponding author due to privacy of patients vaccinated.

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
