# Peer review of "Vaccination Open Day: A Cross-Sectional Study on the 2023 Experience in Lombardy Region, Italy"

_ijerph, 2024, doi:10.3390/ijerph21060685_

Round 1

Reviewer 1 Report

Comments and Suggestions for Authors

This study describes using vaccine Open Days to increase vaccine compliance in Italy Lombardy Region. Getting vaccines or not is an important question nowadays especially after the COVID pandemic. However, there are still some points that would help the study to be designed better.

1.        Please introduce Lombardy region more including more demographic factors that is associated with vaccination, such as age and education. Also, if this is a typical place that somehow represent Italy, please describe. Similarly, how could your study benefit other places in Italy, or Europe places?

2.        Please describe why influenza, anti-COVID and anti-pneumococcal vaccines. Relatively, please describe the pathogens that causes these diseases and briefly talk about the symptoms, which will help you emphasize the importance of taking the vaccines.

3.        Do people take 2 or more vaccines the same day? Any side effects from that? Also, why do a vaccine open day? Probably some people do not trust the vaccines. Talking about the reasons, such as side effects, will make your study more comprehensive. 

4.        Lastly, what are the brand of vaccines and what are the type of vaccines? For instance, COVID vaccines got multiple brands, and each brand has same or different mechanisms. For influenza vaccines, people who are allergic to eggs should be alerted. There are a lot to talk about on the types of vaccines and the different vaccination mechanisms.  

Author Response

Manuscript ID: Ijerph-3000864

Title: Vaccination Open Day: a cross-sectional study on the 2023 experience in Lombardy Region, Italy

Milan, May 20ft, 2024

Dear Editor,

Dear Reviewer,

Thank you for your comments about our submission to Internation Journal of Environmental Research and Public Health.

We carefully considered your precious suggestions and helpful notes, and revised the text accordingly, writing in red the changes made to the manuscript. A detailed list is enclosed.

We hope that the present version of MS will be suitable for publication in Internation Journal of Environmental Research and Public Health.

Sincerely,

Author & co authors

Reviewer 1 OPEN DAY VACC

This study describes using vaccine Open Days to increase vaccine compliance in Italy Lombardy Region. Getting vaccines or not is an important question nowadays especially after the COVID pandemic. However, there are still some points that would help the study to be designed better.

Q1.        Please introduce Lombardy region more including more demographic factors that is associated with vaccination, such as age and education. Also, if this is a typical place that somehow represent Italy, please describe. Similarly, how could your study benefit other places in Italy, or Europe places?

A1.   We have included a table in the results section in order to present the important demographic characteristics in the vaccination settings in the Lombardy region. In discussion, a sub-paragraph was added to explain the importance of the Lombardy settings for vaccination interventions both nationally and internationally.

Q2.        Please describe why influenza, anti-COVID and anti-pneumococcal vaccines. Relatively, please describe the pathogens that causes these diseases and briefly talk about the symptoms, which will help you emphasize the importance of taking the vaccines.

A2.  The decision to administer these three vaccines is due to the season of the vaccination campaign. All three vaccines considered are protective against respiratory diseases that peak at the end of the autumn season and during the winter season. Furthermore, these three vaccines can be co-administered in pairs during the same vaccination session. Moreover, we have added a couple of paragraphs in the introduction in order to describe influenza, COVID and pneumococcal infections, describing both the diseases and the economic and social impact they have. At the same time, we have introduced a sub-section within the discussion to highlight the importance of these vaccinations in the public health sphere.

Q3.        Do people take 2 or more vaccines the same day? Any side effects from that? Also, why do a vaccine open day? Probably some people do not trust the vaccines. Talking about the reasons, such as side effects, will make your study more comprehensive. 

A3.   A maximum co-administration of 2 vaccines in the same session was agreed upon Ministerial Decree and no adverse events were observed during administration. In the discussion we have tried to expand the explanation of the use of a Vaccination Open Day. We have already considered reasons for refusal such as fear of side effects due to previous study performed in the same area. We have also better explained this argument considering the safety of an hospital setting as hypothetical reasons for vaccination acceptance.

Q4.        Lastly, what are the brand of vaccines and what are the type of vaccines? For instance, COVID vaccines got multiple brands, and each brand has same or different mechanisms. For influenza vaccines, people who are allergic to eggs should be alerted. There are a lot to talk about on the types of vaccines and the different vaccination mechanisms.  

A4.  We added a paragraph within the materials to describe both the type and brand of vaccines used and, in the case of different types and brands, the target population of use of each. During anti-COVID and flu vaccinations any allergies to both drugs and food products were investigated in anamnesis.

Reviewer 2 Report

Comments and Suggestions for Authors

The work by Perrone et al is an article presenting the open day event 2023 in Lombardy Region (Italy) for the beginning of influenza and COVID-19 vaccination campaign. The work is well written, references were appropriate. Some revisions are needed to better describe data reported in tables, revise some data included, and clarify some statement that are not always full logical for the reader.  

Please, revise in track changes the manuscript considering:

-        Considering that this is an international journal, the inclusion of the Country of “Lombardy region” in the title can be more informative and immediate to understand.

-        Clarify in the text the meaning of electronic diary created by Fondazione Ca Granda Ospedale Policlinico and the correlation between the cohort of 150 individuals with all subjects recruited during the open day activity.

-        Is it correct in the abstract that 154 vaccines were administered to 150 subjects?

-        Total number of cohorts in Tab 6 and 7 are different. Please clarify or correct.

-        The name of Tab 6 and Tab 7 is identical. I suggest to revise them including some details that differs between tables.

-        Tab 8: in the first column, I suppose that the number “10” refers to October. Please, clarify if needed in the legend or in the figure. Is it correct that the open days were organized in the first 7 days of October for 2021, 2022 and 2023? In the figure legend include the concept that numbers included in the tables refer to number of vaccination.

-        How author can explain the high difference of number of vaccination between 2023 and 2021?

-        Tab. 3 presents data about each ATS booked reservation. Clarify the meaning of reservation with the “availability” and data indicate in its column (number and percentage). Data are difficult to interpret without any description.

-        Tab 7: the total of ATS Milano and Pavia is not the sum of vaccines reported. Revise data updating also the data as percentage.

-        Line 230: include a dot at the end of the sentence.

-        Please, revise the statement at line 264-269 with more clear message.

Comments on the Quality of English Language

 Minor editing of English language required

Author Response

Manuscript ID: Ijerph-3000864

Title: Vaccination Open Day: a cross-sectional study on the 2023 experience in Lombardy Region, Italy

Milan, May 20ft, 2024

Dear Editor,

Dear Reviewer,

Thank you for your comments about our submission to Internation Journal of Environmental Research and Public Health.

We carefully considered your precious suggestions and helpful notes, and revised the text accordingly, writing in red the changes made to the manuscript. A detailed list is enclosed.

We hope that the present version of MS will be suitable for publication in Internation Journal of Environmental Research and Public Health.

Sincerely,

Author & co authors

Reviewer 2

The work by Perrone et al is an article presenting the open day event 2023 in Lombardy Region (Italy) for the beginning of influenza and COVID-19 vaccination campaign. The work is well written, references were appropriate. Some revisions are needed to better describe data reported in tables, revise some data included, and clarify some statement that are not always full logical for the reader.  

Please, revise in track changes the manuscript considering:

Q1.        Considering that this is an international journal, the inclusion of the Country of “Lombardy region” in the title can be more informative and immediate to understand.

A1.   Done

Q2.       Clarify in the text the meaning of electronic diary created by Fondazione Ca Granda Ospedale Policlinico and the correlation between the cohort of 150 individuals with all subjects recruited during the open day activity.

A2.   We have explained the significance of the electronic booking diary in the materials and methods section. We have also better explained the characteristics of the cohort of 150 individuals vaccinated at the Foundation by describing the ratio compared to the general vaccinated population.

Q3.        Is it correct in the abstract that 154 vaccines were administered to 150 subjects?

A3.   Yes, due to the possibility of vaccine co-administration to same patient.

Q4.        Total number of cohorts in Tab 6 and 7 are different. Please clarify or correct.

A4. Corrected

Q5.        The name of Tab 6 and Tab 7 is identical. I suggest to revise them including some details that differs between tables.

A10. Included details about single tables.

Q6.        Tab 8: in the first column, I suppose that the number “10” refers to October. Please, clarify if needed in the legend or in the figure. Is it correct that the open days were organized in the first 7 days of October for 2021, 2022 and 2023? In the figure legend include the concept that numbers included in the tables refer to number of vaccination.

A6. A legend below the table has been introduced to better explain the table itself. No, as indicated in the text, the Open Day was introduced for the first time during the 2023 campaign, making it interesting although very complex to compare the different campaigns.

Q7.        How author can explain the high difference of number of vaccination between 2023 and 2021?

A7. The considerable discrepancy in vaccination rates may be attributed to several underlying factors. As indicated in the text of Table 8, the vaccination open day held in 2023 could have played a role in this variation. This was a new event in the context of previous years, where the campaign did not commence with this specific event. It is also possible that this difference could be due to a different perception of the importance of vaccination between a 'pandemic' and an almost 'post-pandemic' year.

Q8.        Tab. 3 presents data about each ATS booked reservation. Clarify the meaning of reservation with the “availability” and data indicate in its column (number and percentage). Data are difficult to interpret without any description.

A6. A legend below the table has been introduced to better explain the meaning of reservation, availability and availability/1000 columns.

Q9.        Tab 7: the total of ATS Milano and Pavia is not the sum of vaccines reported. Revise data updating also the data as percentage.

A9. Done

Q10.        Line 230: include a dot at the end of the sentence.

A10. Done

Q11.        Please, revise the statement at line 264-269 with more clear message.

A11.   We have completely rewritten the statement to ensure greater clarity and comprehensibility of the message conveyed.

Round 2

Reviewer 2 Report

Comments and Suggestions for Authors

Some minor points need to be addressed:

-        Resentence line 38-40.

-        Define CAP at line 61.

-        Revise spelling at line 131.

-        Revise the name of conjugate vaccine for pneumococcal vaccine.

Comments on the Quality of English Language

Minor editing of English required.